# In Vitro Study of Laser-Assisted Prefabricated Ceramic Crown Debonding as Compared to Traditional Rotary Instrument Removal

**DOI:** 10.3390/ma15103617

**Published:** 2022-05-18

**Authors:** Janina Golob Deeb, Andrew Crowell, Kristen H. Richey, Sompop Bencharit, Caroline K. Carrico, Tiffany L. Williams, Kinga Grzech-Leśniak

**Affiliations:** 1Department of Periodontics, School of Dentistry, Virginia Commonwealth University, Richmond, VA 23298, USA; jgolobdeeb@vcu.edu (J.G.D.); richeykh@vcu.edu (K.H.R.); kgl@periocare.pl (K.G.-L.); 2Department of Pediatric Dentistry, School of Dentistry, Virginia Commonwealth University, Richmond, VA 23298, USA; crowella@vcu.edu (A.C.); twilliams25@vcu.edu (T.L.W.); 3Department of Oral and Craniofacial Molecular Biology, Philips Institute for Oral Health Research, School of Dentistry, Virginia Commonwealth University, Richmond, VA 23298, USA; 4Department of Biomedical Engineering, College of Engineering, Virginia Commonwealth University, Richmond, VA 23284, USA; 5Department of Dental Public Health and Policy, School of Dentistry, Virginia Commonwealth University, Richmond, VA 23298, USA; ckcarrico@vcu.edu; 6Laser Laboratory, Oral Surgery Department, Wroclaw Medical University, 50-425 Wroclaw, Poland

**Keywords:** crown, Er,Cr:YSGG, laser, laser debonding, removal, zirconia

## Abstract

This study compared the laser and rotary removals of prefabricated zirconia crowns in primary anterior and permanent posterior teeth. Sixty-two extracted teeth were prepared for prefabricated zirconia crowns cemented with resin-modified glass-ionomer cement. Specimens underwent crown removals by a rotary handpiece, or erbium, chromium: yttrium-scandium-gallium-garnet (Er,Cr:YSGG) laser. Pulpal temperatures, removal times, and scanning electron microscopy (SEM) examinations were compared. The average crown removal time for rotary and laser methods was 80.9 ± 19.36 s and 353.3 ± 110.6 s, respectively, for anterior primary teeth; and 114.2 ± 32.1 s and 288.5 ± 76.1 s, respectively, for posterior teeth (*p* < 0.001). The maximum temperature for the rotary and laser groups was 22.2 ± 8.5 °C and 27.7 ± 1.6 °C for anterior teeth, respectively (*p* < 0.001); and 21.8 ± 0.77 °C and 25.8 ± 0.85 °C for the posterior teeth, respectively (*p* < 0.001). More open dentinal tubules appeared in the rotary than the laser group. The rotary handpiece removal method may be more efficient than the laser with lower pulpal temperature changes. However, the laser method does not create noticeable tooth or crown structural damage compared to the rotary method.

## 1. Introduction

Recent advancement of laser dentistry expands the applications for lasers from oral surgery, dental implantology, soft tissue surgery, bone and tooth structure surgery and modification, improving wound healing, to removal of restorative materials and restorations [1,2]. Current versatile dental adjustable high-power laser systems provide an efficient and secure method to treat hard tissue and soft tissue as well as allowing debonding of composite materials and removing bonded ceramic restorations [3,4,5,6]. Laser-assisted restorative retrieval or restoration removal utilizes the principle of direct absorption of photon energy by water molecules and residual monomers in luting cements or composite resin materials. The laser absorption of the photons causes the vibration of the water or free monomer molecules in the cements or composite materials, which in turns generates heat energy. The increased energy produces micro-explosions of these water/monomer molecules [6]. This principle of laser function enables sufficient and minimal-invasive hard-tissue preparation [7], as well as composite resin removal and ceramic restoration debonding.

When a significant tooth structure is lost in pediatric patients, often a prefabricated crown such as stainless steel crowns (SSC) or zirconia is prescribed. Most often these teeth are primary anterior teeth and posterior permanent first molars. The SSC have historically been the treatment of choice for restoring carious lesions on primary teeth [8,9,10], with high success rates of 97% [11]. Due to the fast application and good long-term stability, SSCs have also become a treatment option for permanent posterior teeth in children as well as the adult population with special needs [12]. A SSC is often prescribed to preserve tooth structure and integrity for permanent molars with traumatic fractures, large carious lesions, pulp therapy, or developmental defects, such as amelogenesis imperfecta, dentinogenesis imperfecta, and hypomineralized or hypoplastic permanent molars [8,9,10]. While the main advantage of SSCs is the functional durability, the main challenge of SSCs is esthetic concern, especially when used in the anterior esthetic zone [8]. In addition, SSCs have been shown to exhibit corrosion, allow growth of bacterial biofilm, and possibly harbor certain pathological bacteria [13]. More recently, there has been an increase in the use of prefabricated zirconia crowns which offer an esthetically superior alternative to SSCs. Compared to stainless steel, zirconia material decreases plaque accumulation and improves gingival health [14,15]. Prefabricated zirconia crowns have comparable success rates to SSCs in the pediatric population [8]. Shortcomings compared to conventional SSCs include the need for significantly more tooth reduction to provide a passive fit of the crown, resulting in a large cement gap, and a possibility of significant wearing of the opposing dentition compared to SSCs [15]. The esthetic superiority of zirconia crowns has led to increased demand from parents, especially when the anterior dentition has to be restored.

The removal of a crown can sometimes be indicated due to recurrent caries, fractured ceramic, replacement of a prefabricated crown in permanent dentition with a definitive fabricated crown, or inaccurate cementation. While strong cement retention between the crown and tooth is necessary for longevity, the bond often creates a challenge when crown removal is indicated. Multiple instruments and techniques have been used to mechanically remove a crown from the tooth, with the most commonly used technique sectioning a crown with a diamond or carbide bur with a rotary handpiece [16]. Removal of a crown with high-speed handpieces can create an unpleasant experience for pediatric patients due to uncomfortable vibration and loud noises [16]. Distinguishing between the tooth-colored cement and tooth structure can be difficult, thus creating a risk of irreversible unnecessary damage to the underlying tooth structure [16]. Most importantly, crown removal with a high-speed handpiece requires sectioning of the crown rendering the crown unusable.

Prefabricated zirconia crowns can be removed with erbium lasers in an atraumatic manner [17,18]. Laser irradiation offers a predictable method for debonding zirconia crowns, creating minimal to no damage to the tooth or crown surfaces. Laser removal provides a more positive patient experience [19] with less odor, vibration, and noise compared to air rotary instrumentation, and can be used without local anesthetic [20].

The Er,Cr:YSGG laser has an emission wavelength of 2780 nm which is absorbed by chromophores including water, hydrated tissues, residual monomers, and bonding cement containing water [6,21]. Through a process of thermomechanical ablation, laser energy light is transmitted through the ceramic crown and is absorbed by water molecules and residual free monomers within the glass ionomer cement resulting in the disintegration of the cement layer between the crown and the abutment tooth [22]. Zirconia crowns can be retrieved undamaged and can be recemented again if necessary. Studies have shown that laser irradiation can be an efficient and predictable removal method for prefabricated zirconia crowns from permanent and primary molars; however, no studies to date has been conducted on primary anterior teeth or compared the laser method to rotary handpiece removal [17].

The aim of this in vitro study is to analyze and compare the removal time required, intrapulpal temperature change during the procedure, and differences in ceramic and tooth structure integrity for removal of prefabricated zirconia crowns from primary anterior teeth using an Er,CR:YSGG laser and air rotary handpiece.

## 2. Materials and Methods

### 2.1. Specimen Preparation

Extracted teeth were collected in the Virginia Commonwealth University (VCU) Dentistry clinics. The Institutional Review Board was waived for the use of discarded extracted teeth. A sample size of 13–18 in each retrieval group has 80% power to detect an effect size of 1 using a two-group t-test with a 0.05 significance level, assuming that there was a difference between the two methods of at least one standard deviation to be statistically significant. nQuery v.8.7.2 (Statistical Solutions Ltd., San Diego, CA, USA) MTT0-1 was used to estimate effect size based on sample size. Due to a lack of pilot data regarding the expected means and standard deviations, the power analysis was calculated for effect size, which is a function of the means and standard deviation. Eighteen primary anterior and thirteen permanent posterior teeth, deemed restorable with adequate remaining tooth structure, were selected for this study. Extracted teeth were stored as previously described [4].

The abutment teeth were prepared based on the prefabricated zirconia crown manufacturer’s instructions (NuSmile^®^, Houston, TX, USA) using an 850012-C tapered-diamond bur (Premier^®^, Plymouth Meeting, PA, USA) in an Adec TG-97L air-driven handpiece (Henry Schein^®^, Melville, NY, USA), allowing for a minimum of 1 mm of ceramic thickness to ensure a passive seat for a crown [17]. Teeth were numbered consecutively and divided into four groups: Group 1: Anterior Laser retrieval (AL, *n* = 18); Group 2: Anterior Rotary retrieval (AR, *n* = 17); Group 3: Posterior Laser retrieval (PL, *n* = 13); Group 4: Posterior Rotary retrieval (PR, *n* = 13) (Figure 1). Prefabricated zirconia crowns were cemented with designated resin-modified glass ionomer (RMGI), BioCem cement (BioCem Universal BioActive Cement; NuSmile: Houston, TX, USA) [17,23]. Finger pressure was used to stabilize the crown on the tooth during the setting of the cement. Tack curing with a curing light (800–1200 mW/cm^2^) for five seconds on both the buccal surface and 5 s on the lingual surface was performed. Excess cement was removed around the crown margin with gauze followed by curing for 20 s circumferentially for definitive cementation. To prevent desiccation and allow polymerization of the cement, teeth were stored in a specimen container separated by soaked gauze (humidor) in normal saline solution for 48 h prior to crown removal.

### 2.2. Crown Removal Methods

Two different methods were used and compared for crown removal: laser irradiation versus air rotary handpiece removal (Figure 1). Laser crown removal was similar to previous studies [6,17]. Laser irradiation was performed using an Er,Cr:YSGG laser (Waterlase iPlus^®^, Biolase^®^, CA, USA) using settings: 5.0 Watts, 15 Hertz, the fluence in pulse: 39.32 J/cm^2^, power density: 786.3 W/cm^2^, with 50 water and 50 air, with the Turbo handpiece and MX9 laser tip (spot size 900 μm) (Figure 1A). Laser irradiation was performed in 30 s intervals in a sweeping motion on buccal and lingual surfaces. Crown removal was attempted following three minutes of laser irradiation and thereafter every 30 s. Crown removal was achieved by placing gauze over the crown to prevent external damage and performing a gentle rotational movement of the crown using a pair of hemostats (Figure 1C). The total crown removal time was then recorded for each specimen.

The air rotary handpiece removal was achieved using an Adec TG-97L air-driven high-speed handpiece (Henry Schein^®^, Melville, NY, USA) at 400,000 rpm with water spray using a friction grip 850012C Course flame diamond bur (Premier^®^, Plymouth Meeting, PA, USA) (Figure 1B). Crown removal was achieved by cutting the crown in the middle from buccal to lingual surface until it was split in half and fell off the tooth or was able to be removed with a piece of gauze and a pair of hemostats (Figure 1D). The total crown removal time was then recorded for each specimen.

For both crown removal techniques, one operator performed experiments for the anterior and one for the posterior specimens. Each operator was trained to use the laser by an experienced laser clinician. Three specimens for each group were used to calibrate and practice before the experiments were performed. For the posterior samples, a senior (fourth year) dental student performed all experiments. For the anterior samples, a senior (second year) pediatric dentistry resident performed all experiments.

### 2.3. Pulpal Temperature Measurements

Intrapulpal temperature was measured during laser and air rotary handpiece crown removal similar to previous studies [4,17]. A hole was prepared through the root of the tooth directly into the pulp chamber using 330 carbide bur (Henry Schein^®^, Melville, NY, USA). The temperature probe (Sper Scientific^®^ 800008, Scottsdale, AZ, USA) was inserted into the pulpal chamber to measure change in pulpal temperature starting at the baseline and recorded every 30 s throughout the duration of the crown removal process.

### 2.4. Scanning Electron Microscopy Analysis (SEM)

Following crown removal, one specimen from each group underwent scanning electron microscopy (SEM) (JEOL 6610LV, JEOL, Tokyo, Japan) analysis to evaluate the structural integrity and surface of the tooth and zirconia crown following laser irradiation versus high-speed rotary handpiece removal technique.

### 2.5. Statistical Analyses

Differences in time and temperature were compared based on the method of removal (laser vs. rotary handpiece) with equal and unequal variance t-tests, as appropriate. An F-test for equality of variance was used to determine if equal or unequal variance t-test was appropriate. The significance level was set at 0.05 level. SAS EG v.8.2 (SAS Institute, Cary, NC, USA) was used for all analyses.

## 3. Results

### 3.1. Anterior Teeth

Eighteen crowns were removed with a laser and seventeen with the rotary handpiece. The average time (±standard deviation) for removal with the rotary handpiece was 80.9 ± 19.36 s, or approximately 1 min and 20 s. The average time with a laser was 353.3 ± 110.6, or 5 min and 53.3 s. Removal with the laser was significantly longer by an average of 272.4 s (95% CI: 216.8–328.0, *p* < 0.001). Laser removal also had a significantly higher variability (*p* < 0.001) compared to the rotary handpiece removal (Figure 2, Table 1). The maximum temperature recorded and the temperature change during the removal process were also used to compare the two removal methods. The maximum observed temperature with the rotary handpiece was 22.2 ± 0.85 °C compared to 27.7 ± 1.60 °C with the laser. Laser crown removal produced approximately a maximal temperature of 5.5 °C (95% CI: 4.6–6.4), higher than with the rotary handpiece method (*p* < 0.001) (Figure 3). The change in temperature from baseline was recorded during the removal process and was also used to compare the two removal methods. The average temperature change with the rotary handpiece was 0.18 ± 1.16 °C compared to 2.94 ± 1.86 °C with the laser. Laser crown removal produced a temperature change of approximately 2.76 °C (95% CI: 1.69–3.83), higher than with the rotary handpiece (*p* < 0.001) (Figure 4).

### 3.2. Posterior Teeth

A total of 26 posterior crowns were removed with either laser or with a rotary handpiece, 13 in each group. The average time (±standard deviation) for removal with the rotary handpiece was 114.2 ± 32.1 s, or approximately 1 min and 54 s. The average time with a laser was 288.5 ± 76.0 s, or approximately 4 min and 48.5 s. Laser crown removal time was significantly longer by an average of 174.2 s (95% CI: 127.0–221.5, *p* < 0.001) with a higher variability than the rotary handpiece method (*p* = 0.006, Figure 2). The maximum observed temperature with rotary handpiece removal was 21.8 ± 0.77 °C compared to 25.8 ± 0.85 °C with the laser. Laser use for removal was associated with a maximum temperature of 4.0 °C (95% CI: 3.4–4.7), higher than with the rotary handpiece (*p* < 0.0001) (Figure 3, Table 1). The average temperature change with rotary handpiece crown removal was −0.55 ± 1.17 °C compared to 1.78 ± 1.22 °C with the laser method. Laser crown removal had a temperature change of approximately 2.33 °C (95% CI: 1.36–3.30), higher compared to the rotary handpiece method (*p* < 0.001) (Figure 4).

### 3.3. Scanning Electron Microscopy Analysis (SEM)

SEM analysis of the abutment teeth demonstrated rotary cutting streaks on the tooth surface at a low magnification of 500× (Figure 5). At higher magnifications (≥5000×), there were more openings of the dentinal tubules observed in the rotary handpiece removal group compared to the laser group. The dentin surface of the specimen from laser crown removal appeared to have a more intact smear layer and residual cement (Figure 6A,B). Crown specimens from the rotary group presented with more crack and crest lines at higher magnification levels (10,000×), indicating possibly more surface damage of the ceramic material compared to the laser group (Figure 7A,B).

## 4. Discussion

Currently, little information exists in the literature on the examination of laser-assisted crown removal of prefabricated zirconia crowns from primary dentition, nor on comparing the laser crown removal method with an air rotary handpiece technique for prefabricated zirconia crowns. The results from this study suggest that primary zirconia crowns can be successfully removed and reused using laser debonding. However, there are significant differences in the removal time and pulpal temperature change between the laser method and the rotary handpiece method. Previous studies have shown that laser debonding prefabricated zirconia crowns from primary and permanent posterior molars can be successfully accomplished without damaging the underlying tooth structure and preserving the integrity of the crown to be reused if needed [17]. Prefabricated zirconia crowns can be removed by both laser and air rotary handpiece efficiently within six minutes. However, removal times with the laser were significantly longer. While air rotary handpiece removal featured faster retrieval times, this study did not take into consideration the additional time needed for local anesthetic delivery needed for rotary crown removal. In two studies, in vitro debonding of prefabricated zirconia crowns from permanent molars with Er,Cr:YSGG required on average 227 s [17] and 233.1 s [18], which is similar for the posterior teeth in this study, a time of 288 s. The time for primary anterior teeth, an average of 353 s, was, however, much higher than observed for primary posterior teeth (125 s). There may be some anatomical variations of the teeth and possibly operators. Compared to air rotary handpiece crown removal, laser crown removal offers the benefit of delivering pre-emptive analgesia and painless treatment, without the added time and dental fear associated with a needle-injected local anesthetic [24].

Crown removal is a routine dental procedure. It should be noted that removal times for zirconia crowns with an air rotary handpiece can vary greatly depending on the provider’s experience, whereas removal time with the laser is likely dictated by laser settings, cement type and thickness, crown materials, and tooth anatomy/crown retention form [17]. Longer laser removal times may have been related to cement thickness associated with a prefabricated crown that requires a passive fitting concept, where more energy is required to ablate the thicker volumes of cement. Previous studies have shown that greater cement thickness is related to longer debonding time [17]. There are no visible external physical changes in the crown after being subjected to the laser crown removal procedure, thus allowing the reuse of the crown. The crown is destroyed following rotary handpiece removal. The SEM analysis of the ceramic material subjected to laser crown removal appeared intact, with no structural damage similar to previous reports [4,6,17,18]. A downside of the inability to observe changes in crown structure during laser removal is that it is difficult to determine when the crown is ready to be retrieved from the tooth and thus this may reflect in a longer time for crown retrieval.

Thermal stimulation of pulp tissue can occur during crown removal using both the laser and air rotary handpiece techniques. An increase in pulpal temperature by over 5.6 °C is thought to induce pulpal damage [25,26,27] and a pulpal temperature of over 42 °C can cause increased blood flood into the pulp or pulpal hyperemia, resulting in irreversible pulp damage [28]. Increases in pulpal temperature during removal with either the laser or rotary handpiece technique did not exceed 27.7 °C and the temperature change for both methods is less than 2–3 °C. Laser irradiation also showed more variation in temperature and higher temperature readings compared to air rotary removal. Differences in increases in pulpal temperature can be attributed to the functionality of each device. Higher temperature readings associated with laser irradiation can be attributed to longer procedure times, less water spray for cooling, and higher energy needed for crown removal. When crowns are removed with an air rotary handpiece, the energy is absorbed on the surface of the crown and further away from the pulp. Laser energy is absorbed by the underlying chromophore (cement), which is closer in proximity to the pulpal tissue, and thus contributes to greater increases in pulpal temperature during laser crown removal. Primary anterior teeth have a thin layer of remaining dentin that overlies the pulp chamber following tooth preparation, thus the higher temperature change in this group during laser irradiation can probably be attributed to the thin dentinal layer between pulp and luting cement. Adequate water spray is essential to prevent thermal damage to pulpal tissues. Thinner dentin and larger pulpal chambers, especially in pediatric permanent teeth, may be prone to thermal irritation during laser irradiation [29]. The small anatomical primary tooth crown can also be a factor. Thus, it may be preferable to decrease laser setting parameters in the pediatric population, which may lengthen the time required for crown debonding. Pulpal irritation can also be caused by the direct irritation of the dentinal tubules. The SEM analysis of an abutment tooth in the rotary handpiece group revealed greater opening of the dentinal tubules compared to the laser group. The laser removal method appeared to leave a smear layer and cement layer covering the dentin surface that was not apparent in the rotary handpiece group. Greater amounts of open dentinal tubules and lack of a smear layer can result in a higher possibility of post-operative sensitivity following crown removal.

The anterior prefabricated crowns used in this study are more translucent and have lower general physical properties than the posterior crowns. While the rotary removal time appeared to be faster for the anterior crowns than the posterior crowns, reflected by the property of the monolithic zirconia, laser removal times were the opposite. The slower time for laser removal of anterior crowns may be a result of thicker cement layers compared to the posterior crowns. In addition, it is possible that the different thickness of the material may have influenced the removal time. The prefabricated crowns used here are relatively thin, with a maximum thickness of 0.75–0.8 mm, as compared to other studies where the crown thickness was 1–2 mm, or more [4,6,17,18]. This thinner crown material in combination with a thicker uneven cement layer may contribute to the faster removal time for the rotary handpiece group compared to the laser. Note also that the RMGI cement used in this study was very opaque and can be easily distinguished from the dentin, thus contributing to the easy preparation of the rotary handpiece method. Abutment surface area in the case of the definitive customized crown with the uniform thickness of the cement layer was shown to be related to the laser debonding time [17]. Excess cement volume requires longer laser retrieval of the crown [17].

Similar to previous studies, crown removal by laser does not appear to damage the retrieved crowns or the underlying tooth structure [3,6,18,30]. In our study, SEM analysis demonstrated that laser irradiation of zirconia crowns showed no underlying tooth damage and fewer surface irregularities of the crown surface compared to air rotary removal. Evidence of changes in tooth structure at low magnification following removal with a rotary handpiece proves there is a risk of damaging the tooth during crown removal. However, if a crown was to be retrieved due to recurrent caries, then the extent of the caries will impact the dentin surface and smear layer, irrespective of what crown removal technique is employed.

Removal of the crown with a rotary handpiece remains a proven method for crown removal. Erbium laser offers a non-invasive alternative to rotary instruments for ceramic crown removal and can be considered for use in clinical practice. In comparison to traditional bur-based crown removal, laser-assisted crown removal results in an intact crown and tooth after retrieval without the need for local anesthetics. Especially in young and anxious patients, laser-assisted crown removal can be seen as a preferable alternative to conventional bur-based treatment procedures.

This study includes some limitations that should be addressed in future research on ceramic crown removal. First, this study was a benchtop experiment and did not account for the variables that would occur in a clinical practice setting, especially for pediatric patients. Note also that while this study allowed 48 h of cement polymerization in normal saline solution, the intraoral condition may be different. A future study with thermocycling may mimic the oral environment better. Second, the lack of vital pulpal tissues and blood circulation with protective cooling mechanisms present in human subjects can also alter temperature values. Third, future studies should standardize the exact tooth structure remaining following tooth preparation and consistency and standardization in cement thickness should be ensured. In addition, type and size of diamond burs used can have an influence on the removal speed, pulpal temperature, and abutment damage. Fourth, forces used to remove the crowns with the hemostat may have varied depending on cement thickness, size and shape of the tooth, operator strength, and differences in opposing forces that would be present in a human mouth. Finally, future experiments should be performed on human subjects to better replicate intraoral accessibility, in vivo conditions, and various anatomical structures.

## 5. Conclusions

Er,Cr:YSGG laser can be used successfully to retrieve prefabricated zirconia crowns for pediatric patients from both anterior primary teeth and posterior permanent teeth. The retrieval time and temperature generated from laser crown removal are both higher than for the control rotary handpiece method. However, the laser method does not damage crown integrity, allowing the crown to be reused. Furthermore, there was less damage to dentin structure in laser removal compared to the conventional rotary handpiece method.

## Figures and Tables

**Figure 1 materials-15-03617-f001:**
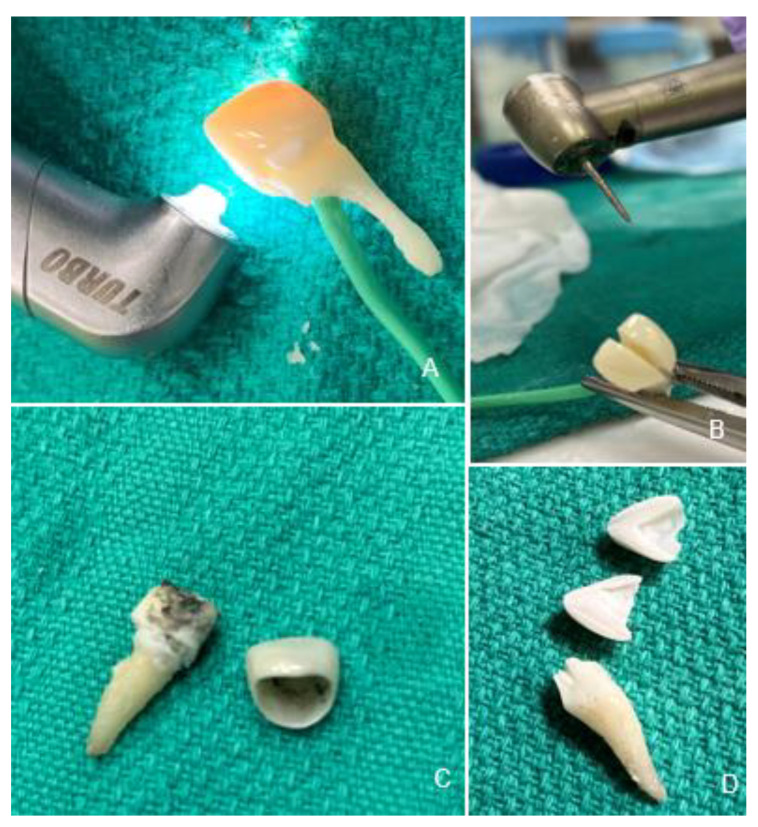
Experimental Procedures. Crown removal with Er,Cr:YSGG laser (**A**), and rotary handpiece (**B**). Laser retrieval results in an intact crown and charred cement covering dentin (**C**). Crown removed with a bur is destroyed with structural damage to the tooth (**D**).

**Figure 2 materials-15-03617-f002:**
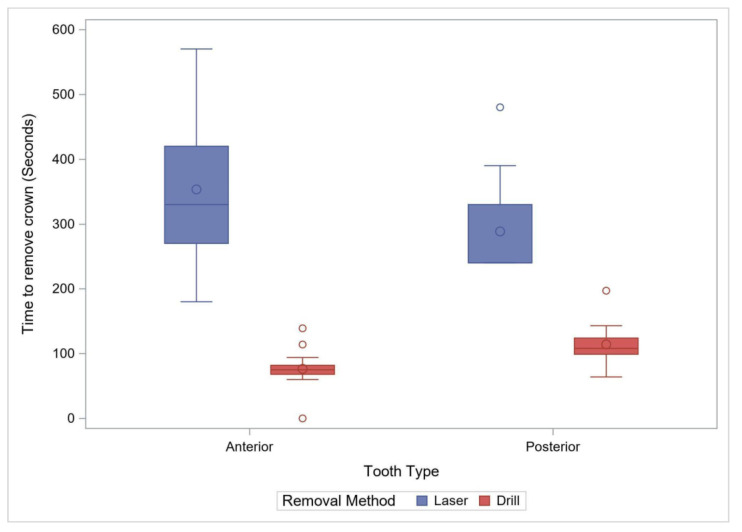
Comparison of Removal Time by Method for Anterior Primary and Posterior Permanent Teeth. The box plots demonstrate 95% confidence interval with “o” within the box plot representing a mean value, and “o” outside the box plot representing an outlier.

**Figure 3 materials-15-03617-f003:**
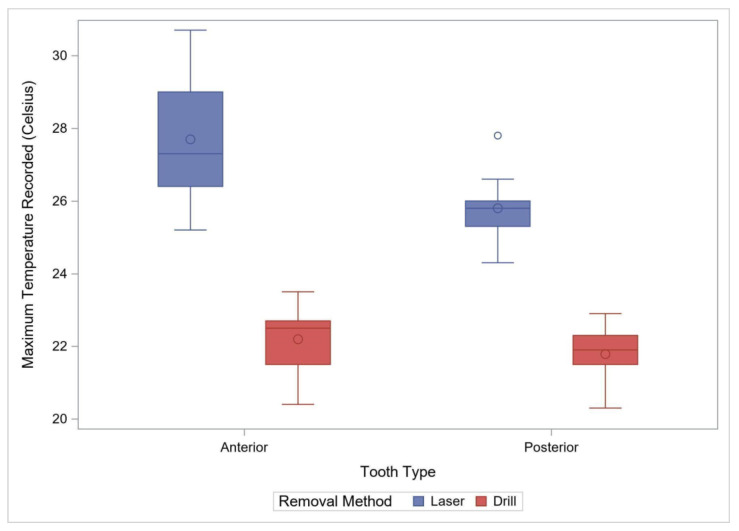
Comparison of Maximum Observed Temperature by Method for Anterior Primary and Posterior Permanent Teeth. The box plots demonstrate 95% confidence interval with “o” within the box plot representing a mean value, and “o” outside the box plot representing an outlier.

**Figure 4 materials-15-03617-f004:**
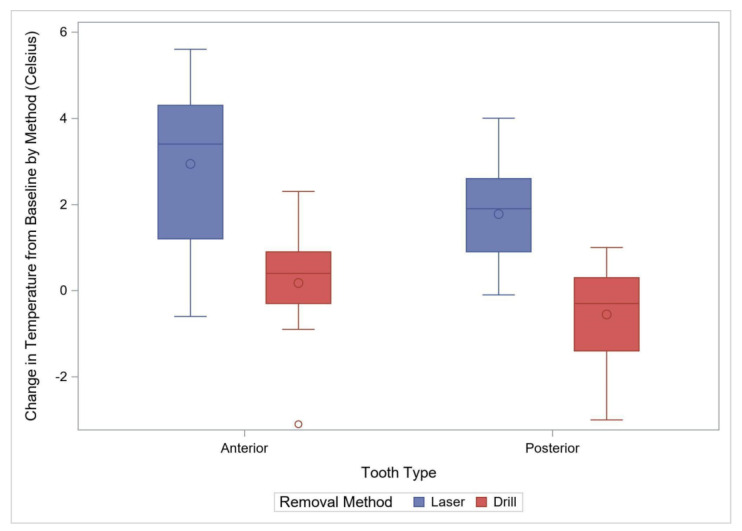
Comparison of Temperature Change from Baselines by Method for Anterior Primary and Posterior Permanent Teeth. The box plots demonstrate 95% confidence interval with “o” within the box plot representing a mean value, and “o” outside the box plot representing an outlier.

**Figure 5 materials-15-03617-f005:**
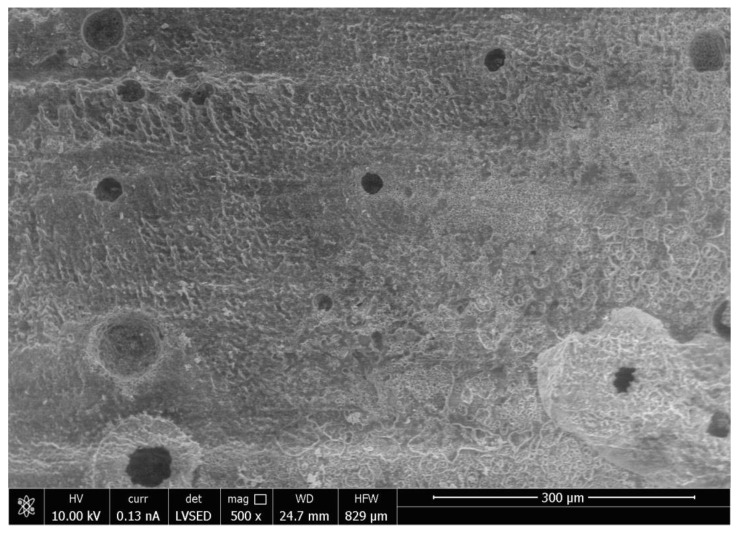
Scanning Electron Microscope Image of the Surface of the Tooth at Magnification 500× Following Crown Removal with Air Rotary Handpiece.

**Figure 6 materials-15-03617-f006:**
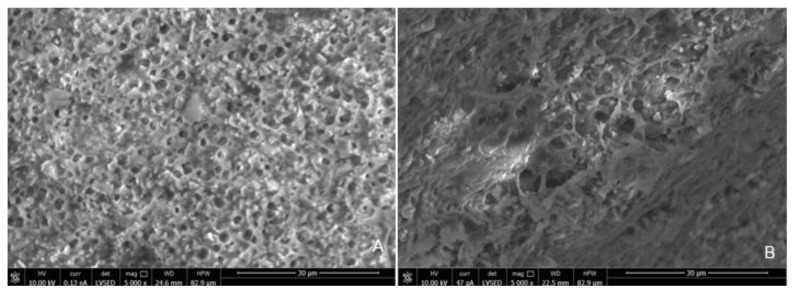
Scanning Electron Microscope Image of the Surface of the Tooth at Magnification 5000× Following Crown Removal with Rotary Handpiece (**A**) and Er,Cr:YSGG Laser (**B**).

**Figure 7 materials-15-03617-f007:**
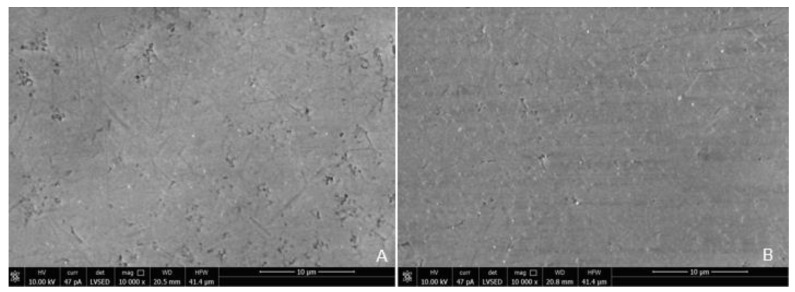
Scanning Electron Microscope Image of the Surface of the Zirconia crown at Magnification 10,000× Following Crown Removal with Rotary Handpiece (**A**) and Er,Cr:YSGG Laser (**B**).

**Table 1 materials-15-03617-t001:** Summary of Debonding Time, Maximum Temperature, and Temperature Change with Laser and Rotary Handpiece Crown Removal *.

	Anterior		Posterior	
	Laser (*n* = 18)	Rotary (*n* = 17)	Laser (*n* = 13)	Rotary (*n* = 13)
Debonding Time (s)	353.3 (110.61)	80.9 (19.36)	288.5 (76.03)	114.2 (32.11)
Size 4 (anterior) ^#^	337.5 (120.76)	83.1 (14.91)		
Size 5 (anterior) ^#^	385.0 (87.81)	79.0 (23.37)		
Maximum Temperature (°C)	27.7 (1.59)	22.2 (0.85)	25.8 (0.85)	21.8 (0.77)
Δ Temperature (°C)	2.9 (1.89)	0.2 (1.160)	1.8 (1.22)	−0.6 (1.17)

* Average (standard deviation). ^#^ The anterior prefabricated crown specimens were used in two sizes, 4 (smaller) and 5 (larger). The posterior specimens were all in the same crown size.

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
