# Peer review of "In Vitro Study of Laser-Assisted Prefabricated Ceramic Crown Debonding as Compared to Traditional Rotary Instrument Removal"

_materials, 2022, doi:10.3390/ma15103617_

Round 1

Reviewer 1 Report

A very interesting study investigating a specific area of restorative management - sectioning of prefabricated zirconia crowns using two techniques. Introduction is clear, discussion is good and addresses the limitations of the study. Certain areas need to be addressed:

  • Lines 126 to 128: Storage in room temp saline is not necessarily replicating clinical presentation in the absence of thermocycling - it certainly allows time for resin conversion/polymerisation and prevents desiccation of specimens. Consider revising/clarifying the statement;
  • Operator details missing (number/experience/calibration, etc.);
  • Table 1 - what is size 4 and 5? Anterior crown size? Also, missing parentheses;
  • The choice of bur can also impact the retrieval time and temp with newer zirconia burs claiming to be more efficient with bonded larger grit size;
  • If a crown was to be retrieved due to recurrent caries, then the extent of the caries will impact the dentine surface and smear layer, irrespective of what crown removal technique is employed;
  • I think the conclusion, while logical, needs to focus on the actual findings on the study (efficiency and temp +structural integrity - rotary performed better in former) and less on advantages of laser use that are outwith the investigation (i.e.: 'safer' to retrieve the crown). Less damage to the pulp is another statement which was only assessed via temp. changes, with rotary performing better (other factors will also play a role). Please revise the conclusion in abstract and main body.

Author Response

Reviewer 1

A very interesting study investigating a specific area of restorative management - sectioning of prefabricated zirconia crowns using two techniques. Introduction is clear, discussion is good and addresses the limitations of the study. Certain areas need to be addressed:

  • Lines 126 to 128: Storage in room temp saline is not necessarily replicating clinical presentation in the absence of thermocycling - it certainly allows time for resin conversion/polymerisation and prevents desiccation of specimens. Consider revising/clarifying the statement;

RESPONSE: We thank the reviewer for pointing this out. 

TEXT CHANGE: The statement was amended as follows.

To prevent desiccation and allow polymerization of the cement, teeth were stored in a specimen container separated by soaked gauze (humidor) in normal saline solution for 48 hours prior to crown removal.”

 An additional limitation statement was added in the Discussion section.

“Note also that while the study allowed 48 hours of cement polymerization in normal saline solution. The intraoral condition may be different. A future study with thermocycling may mimic a better oral environment.”

  • Operator details missing (number/experience/calibration, etc.);

RESPONSE: We appreciate the comment. 

TEXT CHANGE: The following clarifications were added in the Methods section.

For both crown removal techniques, one operator performed for the anterior specimens and one for the posterior specimens. Each operator was trained and certified to use the laser by an experienced laser clinician. Three specimens for each group were used to calibrate and practice before the experiments were performed.”

  • Table 1 - what is size 4 and 5? Anterior crown size? Also, missing parentheses;

RESPONSE: We appreciate the comment.

TEXT CHANGE:  Clarification citation was added in Table 1 as follows.

#The anterior prefabricated crown specimens were used in two sizes, 4 (smaller) and 5 (larger) The posterior specimens were all in the same crown size.”

  • The choice of bur can also impact the retrieval time and temp with newer zirconia burs claiming to be more efficient with bonded larger grit size;

RESPONSE: We appreciate the comment. The reviewer was correct.

TEXT CHANGE: An additional limitation statement was added in the Discussion section.

In addition, type and size of diamond burs used can have an influence in the removal speed, pulpal temperature, and abutment damage.”

  • If a crown was to be retrieved due to recurrent caries, then the extent of the caries will impact the dentine surface and smear layer, irrespective of what crown removal technique is employed;

RESPONSE: We agreed with the reviewer.

TEXT CHANGE: An additional  statement was added in the Discussion section.

However, if a crown was to be retrieved due to recurrent caries, then the extent of the caries will impact the dentine surface and smear layer, irrespective of what crown removal technique is employed.”

  • I think the conclusion, while logical, needs to focus on the actual findings on the study (efficiency and temp +structural integrity - rotary performed better in former) and less on advantages of laser use that are outwith the investigation (i.e.: 'safer' to retrieve the crown). Less damage to the pulp is another statement which was only assessed via temp. changes, with rotary performing better (other factors will also play a role). Please revise the conclusion in abstract and main body.

RESPONSE: We agreed with the reviewer.

TEXT CHANGE: The conclusion statement in the abstract and the main text are now read.

In the Abstract:

The rotary handpiece removal method may be more efficient than the laser with lower pulpal temperature changes. However, the laser method does not create noticeable tooth  or crown structural damage compared to the rotary method.”

In the Conclusions:

Er,Cr:YSGG laser can be used successfully to retrieve prefabricated zirconia crowns for pediatric patients from both anterior primary teeth and posterior permanent teeth. The retrieval time and temperature generated from laser crown removal are both higher than the control rotary handpiece method. However, the laser method does not damage the crown integrity allowing the crown to be reused. Furthermore, there was less damage to dentin structure in laser removal compared to the conventional rotary handpiece method.“

Reviewer 2 Report

Dear. Authors,

This topic was evaluated with removal method of zirconia crown using an Er,CR:YSGG laser and air rotary handpiece was very interesting for researchers and pediatric clinicians.

There are several issues that should be addressed in the manuscript before further consideration for publication.

  1. Materials and Method, Line 106.

Number of specimens is determined by effect size, power, difference between two methods and standard deviation.

 You should add software or calculation formula with some references in manuscript.

2.2 Crown removal methods, Line 134.

Two different methods are used for crown removal.

How many operators did you use in this study?

Number of operators and years of their clinical experience should be mentioned in manuscript.

Results, 173-202.

Replace “p-value ” with “p” in manuscript.

Line 173-174.

The average time and standard deviation are mentioned in results and figures.

However, there are median and quartiles in figures.

You should modify manuscript or figures.

Discussions, Line 302-310.

Removal time is influenced by Crown thickness.

I think removal time is influenced by not only thickness but also translucency of zirconia material.

You should add in discussion if you have any information about NuSmile®

Line 86 and others,

and can be used without local anesthetic.[20]

Add a “period” after reference No.

Author Response

REVIEWER #2

This topic was evaluated with removal method of zirconia crown using an Er,CR:YSGG laser and air rotary handpiece was very interesting for researchers and pediatric clinicians.

There are several issues that should be addressed in the manuscript before further consideration for publication.

  1. Materials and Method, Line 106.

Number of specimens is determined by effect size, power, difference between two methods and standard deviation.

 You should add software or calculation formula with some references in manuscript.

RESPONSE: We appreciate the comment. 

TEXT CHANGE: The information on the software and version was added.  

"nQuery v.8.7.2 (Statistical Solutions Ltd., San Diego, CA) MTT0-1 was used to estimate effect size based on sample size. Due to a lack of pilot data regarding the expected means and standard deviations, the power analysis was calculated for effect size which is a function of the means and standard deviation."

2.2 Crown removal methods, Line 134.

Two different methods are used for crown removal.

How many operators did you use in this study?

Number of operators and years of their clinical experience should be mentioned in manuscript.

RESPONSE: We appreciate the comment. 

TEXT CHANGE: A clarification was added in the Methods section.

 “For both crown removal techniques, one operator performed for the anterior specimens and one for the posterior specimens. Each operator was trained and certified to use the laser by an experienced laser clinician. Three specimens for each group were used to calibrate and practice before the experiments were performed. For the anterior samples, a senior (fourth year) dental student performed all experiments. For the posterior samples, a senior (second year) pedodontic resident performed all experiments.”

Results, 173-202.

Replace “p-value ” with “p” in manuscript.

 RESPONSE: We appreciate your comment.

TEXT CHANGE:  All “p-value” are now “p”.

Line 173-174.

The average time and standard deviation are mentioned in results and figures.

However, there are median and quartiles in figures.

You should modify manuscript or figures.

RESPONSE: We appreciate the comment. However, we would like to illustrate the figures and tables/data description to demonstrate the whole characteristics of the data without too much redundancy. That was the reason why additional information (medians, interquartiles) are present in the figures.

TEXT CHANGE: N/A 

Discussions, Line 302-310.

Removal time is influenced by Crown thickness.

I think removal time is influenced by not only thickness but also translucency of zirconia material.

You should add in discussion if you have any information about NuSmile®

RESPONSE: We truly appreciate this kind insight. 

TEXT CHANGE: An additional discussion was added.

The anterior prefabricated crowns used in this study are more translucent and have lower general physical properties than the posterior one. While the rotary removal time appeared to be faster for the anterior crowns than the posterior ones reflecting in the property of the monolithic zirconia. The laser removal times appeared to be opposite. The slower time for laser removal of anterior crowns may be a result of thicker cement layers compared to the posterior crowns. In addition, it is possible that different thickness of the material may have influenced the removal time.”

Line 86 and others,

and can be used without local anesthetic.[20]

Add a “period” after reference No.

RESPONSE: Truly appreciate the comment.

TEXT CHANGE: The “period” is now placed after reference citation.

Reviewer 3 Report

The article "In-vitro Study of Laser-Assisted Prefabricated Ceramic Crown debonding as compared to Traditional Rotary Instrument Removal" is potentially interesting, but, in my opinion, it's not significant enough for Materials. A better argumentation of the subject would be of interest, in the Introduction part.

Punctual comments:

Remove sections from the abstract.

Line 38. Please do not resume the recent advances of lasers in dentistry only to removal of restorations and crowns. 

Lines 49-59. Please better place the information regarding SSC, in the context of  your paper. 

Please explain the reason of selecting anterior primary teeth and posterior permanent ones. 

Line 188, 202. Please remove "see", give the figure in brackets.

Place the figure within the text, when mentioned, not later.

Same for the table. Mention the table in the main text.

Please use the proper format for the references.

Author Response

REVIEWER #3

The article "In-vitro Study of Laser-Assisted Prefabricated Ceramic Crown debonding as compared to Traditional Rotary Instrument Removal" is potentially interesting, but, in my opinion, it's not significant enough for Materials. A better argumentation of the subject would be of interest, in the Introduction part.

Punctual comments:

Remove sections from the abstract.

RESPONSE: We appreciate your comment.

TEXT CHANGE: Sections were removed from the abstract. The abstract is now read.

This study compared the laser and rotary removals of prefabricated zirconia crowns in primary anterior and permanent posterior teeth. Sixty-two extracted teeth were prepared for prefabricated zirconia crowns cemented with resin-modified glass-ionomer cement. Specimens underwent crown removals by a rotary handpiece; or erbium, chromium: yttrium-scandium-gallium-garnet (Er,Cr:YSGG) laser. Pulpal temperatures, removal times, and scanning electron microscopy (SEM) examinations were compared. The average crown removal time for rotary and laser methods were 80.9±19.36 sec; and 353.3±110.6 sec, respectively for anterior primary teeth; and 114.2±32.1 sec and 288.5±76.1 sec, respectively for posterior teeth (p<0.001). The maximum temperatures for the rotary and laser groups were 22.2±8.5°C and 27.7±1.6°C for anterior teeth, respectively (p<0.001); and 21.8±0.77°C and 25.8±0.85°C for the posterior teeth, respectively (p<0.001). More open dentinal tubules appeared in the rotary than the laser group. The rotary handpiece removal method may be more efficient than the laser with lower pulpal temperature changes. However, the laser method does not create noticeable tooth  or crown structural damage compared to the rotary method.”

Line 38. Please do not resume the recent advances of lasers in dentistry only to removal of restorations and crowns. 

RESPONSE: We thank the reviewers for the comment.

TEXT CHANGE: The sentence is expanded to cover more range of laser applications.

Recent advancement of laser dentistry expands the applications for lasers from oral surgery, dental implantology, soft tissue surgery, bone and tooth structure surgery and modification, improving wound healing, to removal of restorative materials and restorations [1,2].”

Lines 49-59. Please better place the information regarding SSC, in the context of  your paper. 

RESPONSE: Thank you for pointing out this disconnection.

TEXT CHANGE: A linking sentence was added as follows.

When a significant tooth structure is lost in pediatric patients, often a prefabricated crown such as stainless steel crowns (SSC) or zirconia is prescribed.”

Please explain the reason of selecting anterior primary teeth and posterior permanent ones. 

çThank you for pointing this out.

TEXT CHANGE: An additional sentence linking the prefabricated crowns to the most common affected teeth was added.

Most often these teeth are primary anterior teeth and posterior permanent first molars. “

Line 188, 202. Please remove "see", give the figure in brackets.

RESPONSE: Thank you for the comment.

TEXT CHANGE: All “See Figure”s are now changed to “(Figure…).

Place the figure within the text, when mentioned, not later.

Same for the table. Mention the table in the main text.

Please use the proper format for the references.

RESPONSE: Truly appreciate the comment.

TEXT CHANGE: All Figures are now closer to text when mentioned. Table is now cited in the Results section. The references are in the proper format now.